# Model to Predict Healthcare Behaviors: Comparison of a Chilean and Mexican Sample

**DOI:** 10.3390/ijerph191610067

**Published:** 2022-08-15

**Authors:** María José Rivera Baeza, Natalia Salinas-Oñate, Daniela Gómez-Pérez, Rolando Díaz-Loving, Manuel S. Ortiz

**Affiliations:** 1Post-Doctorate Unit of Psychosocial Studies, Faculty of Psychology, Universidad Nacional Autónoma de México, Mexico City 04510, Mexico; 2Academic of the Department of Psychology, Faculty of Health Sciences, Universidad Católica de Temuco, Temuco 4813302, Chile; 3Academic of the Department of Psychology, Faculty of Education, Social Sciences and Humanities, Universidad de La Frontera, Temuco 4811322, Chile; 4Academic of the Psychosocial Studies Unit, Faculty of Psychology, Universidad Nacional Autónoma de México, Mexico City 04510, Mexico

**Keywords:** culture, beliefs, physicians, mistreatment, emotions

## Abstract

(1) Background: Adherence to treatment and medical check-ups are important for health outcomes, but low adherence to treatment is a common phenomenon. Thus, we aimed to examine the role of cultural beliefs about physicians, perceived mistreatment, and emotions associated with the experience of mistreatment as an antecedent of healthcare behavior among Chilean and Mexican primary care patients using Betancourt’s model for the study of health behavior. (2) Methods: This is a multivariate cross-sectional study with a non-probabilistic sample of 326 Mexican and 337 Chilean participants. Multigroup structural equation modeling was used to test the structural relations among the cultural and psychological variables as determinants of healthcare avoidance behavior. (3) Results: The results revealed a pattern of associations that work in the same way for Chilean and Mexican samples. Negative cultural beliefs about physicians have a direct effect on avoidance behaviors in healthcare. In addition, this effect is mediated through psychological factors, such as perception of mistreatment and negative emotions associated with mistreatment. (4) Conclusions: A structural invariance test showed that the perception of mistreatment and negative emotions were less intense for Chileans than Mexicans. In contrast, the association between negative emotions and avoidance behaviors was less intense for Mexicans.

## 1. Introduction

In response to changes in the global epidemiological profile and the increase in non-communicable diseases affecting countries all over the world, particularly in Latin America and the Caribbean [1], the World Health Organization (WHO) has proposed to promote universal health coverage [2] to increase access to quality health services and meet various needs of the population and emerging challenges [3,4]. Although progress has been made towards these objectives proposed by the WHO, i.e., the focus of Latin American policymakers on increasing health coverage among poor individuals [5], people living in unfavorable social conditions are still exposed to greater risk factors and have inadequate access to health services [6]. Cultural and psychological factors also determine health behaviors [6,7,8,9] that can favor or hinder the achievement of the goals proposed by the WHO.

Evidence shows that socially shared beliefs about physicians influence health behaviors, such as the discontinuation of treatment in women diagnosed with cancer, the postponement of preventive checkups, and the delay of medical consultations [7,10]. These beliefs are also associated with perceptions of discrimination and ill treatment in health care, negatively impacting behaviors such as non-adherence to treatment [11], non-attendance at medical appointments [12], reduced willingness to make use of preventive healthcare services [12,13,14,15], delays in seeking care and medications [11,13,14,15], and the replacement of conventional health care by alternative medicine [16].

It has been shown that perceived discrimination or healthcare mistreatment (in this research, the terms ‘discrimination’ and ‘mistreatment’ will be used interchangeably; see (38) for more information) generates negative emotions, such as anger, sadness, and anxiety, that can directly affect health behaviors. Thus, negative emotions could act as mediating variables in the relationship between perceived discrimination in health care and the discontinuation of treatment [10].

Several theoretical models have been developed or adapted to explain health behaviors. Betancourt’s integrative model [7,16] has been successful in predicting healthcare-seeking behaviors and the continuity of care. This model, which guides the present study, proposes a structure of relationships that incorporate sources of cultural variation, as well as cultural and psychological factors, as predictors of health behaviors, as is shown in Figure 1. The model defines culture as the beliefs, norms, expectations, roles, and values that are socially shared by a group and may be relevant to achieving an improved understanding of health behaviors [17]. Cultural elements are a function of population categories (A), such as ethnicity, gender, or socioeconomic level. They are considered the most distal behavioral determinants and a source of cultural variation. Cultural factors, such as negative beliefs about physicians (B), are unique to the sample of interest and connected to psychological factors, such as perceived discrimination, motivation, and emotions, among others (C). Cultural factors are also the most proximal determinants of health behavior (D). Therefore, it is possible to identify how culture affects health behavior directly or indirectly through psychological processes (See Figure 1).

Betancourt’s model has been used to study the impact of health inequities. The literature shows that beliefs, which generally originates from previous experiences with healthcare professionals, affect interactions between healthcare professionals and patients [18], and if patients perceive discrimination and mistreatment in healthcare services, they tend to exhibit lower therapeutic adherence to the treatments [17] and a lower willingness to use health services [7,13,14,15,19].

A study carried out in Chile used the model mentioned above to analyze the relationships between negative cultural beliefs about physicians, perceived discrimination, and negative emotions associated with discrimination as predictors of several health behaviors [20] and showed that negative cultural beliefs about physicians were related to lack of motivation to return for checkups, the rejection of professionals, and neglect of one’s health. In addition, a mediation mechanism was observed through psychological processes whereby negative cultural beliefs about physicians were associated with perceived discrimination and negative emotions in patients, which led to avoidance behaviors in health care [10]. This is relevant because the evidence indicates that the healthcare provided by physicians is an even more essential component of satisfaction with health services than the infrastructure or facilities [21].

Although this model has been used in the United States and Chile, there is a gap in the literature with respect to whether the proposed associations are invariant regarding the cultural context. Thus, we aimed to compare this model using a Chilean and a Mexican sample, considering that these two countries face similar challenges in terms of health issues, such as reducing mortality and morbidity rates and improving people’s health [4]. There are also cultural variations between the two countries regarding their beliefs. Among Mexicans, cultural beliefs about physicians are predominantly positive, especially those associated with the idealization of the professional as a paternalistic and family-friendly character who seeks the well-being of the patient [9]; in Chile, positive aspects are related to the interaction, highlighting attributes such as trustworthiness and confidence.

On the other hand, the investment incurred by each country for health care differs considerably, with Mexico allocating 6.3% of its gross domestic product to health and Chile allocating 7.8%, with a per capita expenditure on health of USD 677 in Mexico and USD 1137 in Chile [22]. Despite differences in economic and health macroindicators, the two countries face similar challenges, such as ensuring that protection and access to health services are of quality to contribute to social equity, especially for vulnerable groups. This is even more relevant, considering that these challenges are related to the goals set out in the 2030 Agenda for Sustainable Development of the United Nations and by the guidelines of the WHO [23].

Given that one of the essential elements for the quality of care is the perception of good treatment by professionals (e.g., physicians), this study was proposed as follows to compare a relationship model of healthcare behaviors that includes the antecedent factors of cultural beliefs about physicians, perceived mistreatment in health care, and emotions associated with the experience of mistreatment in Chile and Mexico. Thus, we aimed to examine the role of cultural beliefs about physicians, perceived mistreatment, and emotions associated with the experience of mistreatment as an antecedent of healthcare behavior among Chilean and Mexican primary care patients. Specifically, the hypotheses of this study sought to assess the role of negative beliefs about physicians, with an expectation that they would be directly associated with avoidance behaviors in health care and have a direct effect on perceived mistreatment. Furthermore, perceived mistreatment is expected to be directly associated with avoidance behaviors in health and associated with discrimination through negative emotions.

## 2. Materials and Methods

### 2.1. Participants

A non-probabilistic, purposive sampling approach was used to select 663 participants. The inclusion criteria for participants in this study were the following: (1) older 18 years of age; (2) receiving medical care in the private or public healthcare system; (3) at least one encounter with a healthcare provider during the last year. Participants who presented limitations in answering the instruments, such as visual impairment, were excluded. Of the participants, 50.8% were Chilean, and 49.2% were Mexican, and the majority were women (85.4% in Chile and 64.9% in Mexico).

### 2.2. Instruments

*Cultural beliefs about physicians*: The subscale of negative beliefs about health professionals [11] was used. This subscale includes 13 items that evaluate the degree of agreement with a statement (e.g., “Doctors treat their patients badly”). It uses a 5-point Likert scale (*1* = *strongly disagree* to *5* = *strongly agree*). The higher the score, the more negative the belief about physicians. This scale presents adequate psychometric properties for Chile [11] and Mexico [24]. The scale presented good reliability for both samples (Chile α = 0.96 and Mexico α = 0.88).

*Mistreatment or unfair treatment in health*: The Baeza-Rivera unfair treatment scale [25] was used. It contains 14 items that present various situations of unfair treatment in encounters with health professionals (e.g., “You have been treated worse than other people”). The participants were asked to respond about how frequently they have experienced a negative encounter in healthcare settings, with options ranging from 1 = *never* to 5 = *always*. A higher score on the scale indicates a higher perception of unfair treatment. Excellent reliability indices were observed in both samples (Chilean sample, α = 0.90; Mexican sample, α = 0.89).

*Emotions and affective states associated with health mistreatment*: The Baeza-Rivera scale of emotions and affective states [25] was used. It contains 13 items that reflect the intensity of emotions and affective states resulting from the reported mistreatment (e.g., “anger”). The response format is a 5-point scale (1 = *not at all* and 5 = *very much*). A unifactorial structure and excellent reliability indices were observed in both the Chilean and the Mexican samples in the analyses of psychometric properties (Chilean sample, α = 0.97; Mexican sample, α = 0.94).

*Health behaviors*: The avoidance behavior subscale of the Scale of Reactions to Experiences of Unequal Treatment in Health [26] was used. This subscale consists of seven items that assess the degree of agreement with possible avoidance behaviors (e.g., “If I could, I would go to another health center”). It uses a 5-point Likert scale, ranging from 1 = *strongly disagree* to 5 = *strongly agree*. Good reliability indices for both samples were observed (Chilean sample, α = 0.90; Mexican sample, α = 0.77).

In addition, the participants indicated their age, gender, level of education, marital status, and socioeconomic class. To review comparison between the variables by sex (Appendix A).

### 2.3. Procedure

The instruments were administered in Spanish. To ensure the correct interpretation of the items, the scales were adapted for the linguistic comprehension of the Mexican sample. For this purpose, a panel of disciplinary and methodological experts was convened to examine the items and evaluate their semantic equivalence, cultural relevance, and linguistic adaptation.

In both cases, the procedure was conducted in accordance with the ethical principles of the Declaration of Helsinki and the American Psychological Association. The Ethics Committee of Servicio de Salud Araucanía Sur (Res. Nº 1179, 6 March 2014) approved the project. Participants who were eligible for inclusion were contacted in various public places (e.g., squares and health centers) by the researcher in charge and trained students. They were informed about the study’s objective, and those who decided to participate signed an informed consent form explaining the procedure and the ethical safeguards for this type of research. The questionnaire was presented in a physical format and took approximately 15 min to complete.

### 2.4. Statistical Analyses

The data were analyzed with STATA 14.0 software. Descriptive analyses were performed to characterize the sample, compare means, and perform a multigroup structural equation modeling analysis (SEM). The proposed relationship pattern was modeled by imposing successive restrictions using the method proposed by Byrne [27]. First, the model was adjusted separately for the Chilean and Mexican samples. Subsequently, a configural model was established and used as a reference model for the following comparisons. Then, the metric equivalence (factor loading restriction) and structural invariance (path restrictions) were estimated. For each comparison, a chi-square likelihood test (LRT ∆χ^2^) and the difference in the absolute value of CFI (∆CFI > 0.01) [28] were used. The Wald test was used to test differences between paths. The overall goodness of fit of the model was estimated with the following indicators: non-significant chi-square (χ^2^), comparative fit index (CFI > 0.95), RMSEA (<0.08) [27], Tucker–Lewis Index (TLI > 0.95), and SRMR (<0.05) [29].

The analyses were performed with Satorra–Bentler correction because multivariate normality was not assumed (*p* < 0.05).

## 3. Results

### 3.1. Sociodemographic Characteristics and Comparison between Groups

Table 1 shows the differences between Mexican and Chilean participants regarding sociodemographic variables. The results show statistically significant differences with small effect sizes in all variables considered: gender, age, marital status, socioeconomic class, and level of education. In both samples, most participants were women (64.9% in the Mexican sample and 85.4% in the Chilean sample). The average age was 31.02 years in the Chilean sample and 32.89 years in the Mexican sample. Regarding marital status, the majority in the Mexican sample were single (50.5%), and in the Chilean sample, the majority was married (44.7%). Most Mexicans (64.9%) and Chileans (51.7%) indicated that they belong to the middle socioeconomic level. Regarding the level of education, most Chilean participants had a university education (47.8%), whereas most Mexican participants had a secondary education (56%) (Table 1).

### 3.2. Association among Variables of Interest

A correlation analysis was performed between the variables of the model, as shown in Table 2. Significant and positive correlations were observed between negative beliefs about physicians and demotivation (*r* = 0.307), delayed medical care (*r* = 0.248), and refusal of help from healthcare providers (*r* = 0.291), whereas the relationship between perceived mistreatment and the indicators of care avoidance was more pronounced, especially demotivation (*r* = 0.417), delayed medical care (*r* = 0.402), and refusal of help from healthcare providers (*r* = 0.403); see Table 2.

### 3.3. Multigroup Structural Equation Modeling

*Test of the hypothesized model*. This analysis was performed for each sample separately and without any restrictions on the model. The results show an excellent fit to the data from both analyzed samples (Chilean sample [χ^2^_(15)_ = 13.914, *p* = 0.532, RMSEA = 0.000 [0.000–0.076], CFI = 1.000, TLI = 1.000, SRMS = 0.030]; Mexican sample [χ^2^_(15)_ = 8.883, *p* = 0.884, RMSEA = 0.000 [0.000–0.028], CFI = 1.000, TLI = 1.000, SRMR = 0.015]).

*Test of configural invariance*. Because the model presented an excellent fit for both samples, we tested a model with the country as a grouping variable (Chile or Mexico). The purpose of this analysis was to demonstrate that in both samples, each latent factor has the same indicators. Therefore, no restrictions were imposed, and an excellent overall fit was observed (χ^2^_(30)_ 23.471, *p* = 0.795, RMSEA = 0.000 [0.000–0.042], CFI = 1.000, TLI = 1.000, SRMR = 0.024). The obtained results were used as a reference for comparison with the following models.

*Measurement invariance test*. In this step, we intended to determine that the latent factors have the same semantic meaning in both samples. Therefore, equality restriction was imposed on all factorial loadings of the latent factors, making the coefficients invariant for both Chileans and Mexicans. The fit indicators were excellent (χ^2^_(34)_ 32.390, *p* = 0.547, RMSEA = 0.000 [0.000–0.041], CFI = 1.000, TLI = 1.000, SRMR = 0.026). When comparing this more restrictive model with the less restrictive (configural) model, no statistically significant differences were found (∆χ^2^_(4)_ 32.390, *p* = 0.063), leading us to conclude that the measurement model operates in the same way for Chileans and Mexicans and that the factors have the same meaning for both groups.

*Test of structural invariance*. Because the previous steps showed that the theoretical model has the same form and the latent factors have a similar meaning in both samples, an equality restraint in the structural paths was imposed in this step. The goodness-of-fit indicators for this more restrictive model were very good (χ^2^_(35)_ 51.625, *p* = 0.035, RMSEA = 0.042 [0.012–0.065], CFI = 0.992, TLI = 0.987, SRMR = 0.071). However, after comparing this model with the reference (configural) model, we observed statistically significant differences between them (∆χ^2^_(4)_ 28.154, *p* < 0.001), leading us to conclude that there are differences in at least one of the paths. Thus, we ran a Wald test, identifying differences in the path from mistreatment to negative emotions, where the magnitude was lower in the Chilean sample (*β* = 0.514, *p* = 0.001) than in the Mexican sample (*β* = 0.681, *p* = 0.001). In the path between negative emotions and avoidance behavior in healthcare, the magnitude was larger in the Chilean sample (*β* = 0.670, *p* = 0.001) than in the Mexican sample (*β* = 0.561, *p* = 0.001) (Table 3).

*Test of research hypotheses*. The structure of associations proposed in this study posits that negative beliefs about physicians can be directly associated with avoidance behaviors in health care and, indirectly, through perceived mistreatment. Furthermore, perceived mistreatment was expected to influence avoidance behaviors in health care through negative emotions associated with perceived discrimination. We found that the proposed model works for both the Chilean and the Mexican samples. Figure 2 shows the indicators, loadings, and regression coefficients for Chilean and Mexican participants (in bold and italics, respectively). The results show a direct relationship between negative beliefs about physicians and avoidance behaviors in health care, with a larger magnitude for Chileans than Mexicans. There is also a positive relationship between negative beliefs about physicians and perceived mistreatment, with a larger magnitude for the Chileans. For both Chileans and Mexicans, there is no statistically significant relationship between perceived mistreatment and avoidance behavior in health care. However, there is a direct correlation between perceived mistreatment and negative emotions associated with perceived discrimination, with a larger magnitude for the Mexican sample. Another direct relationship exists between negative emotions and avoidance behaviors in health care, with a larger magnitude for the Chilean sample.

## 4. Discussion

Apart from human, economic, and technological resources, another challenge in healthcare is to safeguard the quality of care, given its relevance to the behaviors and treatment adherence of patients. Given the importance of cultural (beliefs about physicians) and psychological (perceived mistreatment and associated negative emotions) variables in health care behaviors, in this study, we aimed to compare a relationship model of healthcare behaviors that includes cultural beliefs about physicians, perceived mistreatment in health, and emotions associated with the experience of mistreatment as antecedent factors in a Chilean and a Mexican sample.

Concerning the hypotheses of the study, the results shows that healthcare behaviors in both countries can be explained by a similar structure of relationships between cultural and psychological variables. Negative cultural beliefs about physicians are associated with decreased motivation regarding treatment adherence, postponing subsequent medical encounters, or rejecting the professional, regardless of nationality. In addition, we observed that when negative cultural beliefs about physicians are greater, there is an increased perception of unfair treatment or mistreatment, generating emotions such as anger, anxiety, and sadness, which impact the negative consequences in health care [20]. Furthermore, as in other studies, we observed that perceived mistreatment did not have a direct effect on the behaviors but was mediated by negative emotions, i.e., emotions could promote healthcare behaviors, as they are relevant in the interactions between healthcare providers and patients and may be influenced by cultural characteristics [6,20].

The results indicate that the pattern of relationships is invariant and works in the same way for Chileans and Mexicans. The observed differences are only present in the magnitudes of associations, especially in the effect of perceived mistreatment experiences on negative emotions, which is more intense in Mexico than in Chile. In contrast, the effect of negative emotions on avoidance behavior consequences is stronger in Chile than in Mexico, which means that the experiences of mistreatment led to more negative emotions in the Mexican than the Chilean participants. However, the Chilean participants experienced more negative consequences associated with negative emotions.

The above could be due to specific cultural characteristics of the Mexican participants based on historic sociocultural premises [30] in which they are defined as people with respect for and fear of authority, with a greater acceptance of traditional premises, which involve stronger external control; this means that they show greater obedience to authority and seek kindness pleasantry; their emotions are built on family harmony and values, with a strong emphasis on obedience, with the notion of love being more important than power and with considerable fear of authority; happy emotions are openly displayed, but negative emotions are internalized so as not to affect relationships [30,31,32]. In this context, physicians are seen as health authorities, so it is likely that perceived mistreatment and negative emotions do not necessarily lead to behaviors that are detrimental to health.

These results highlight the importance of cultural variables in the medical care process in the Chilean and Mexican contexts, especially regarding negative cultural beliefs about physicians, which are difficult to modify and contribute to negative stereotypes after being socially shared. Moreover, socially shared beliefs are directly related to sustained interactions with a particular group (e.g., healthcare professionals) [33] to the extent that when patients collectively experience more positive encounters, their beliefs about physicians are more favorable and, according to the literature, more positive behaviors and consequences for their health could be expected. Thus, a reduction in perceived mistreatment by physicians could have a positive impact on health outcomes.

Mistreatment or discrimination not only violate an ethical commitment but also accentuate existing differences and inequities. Various studies have document that the perception of empathy in professionals improves the continuity of care and reduces the probability of experiencing negative emotions in the encounters. When patients evaluate clinical encounters more positively and experience positive emotions, they continue their checkups [6,7,20] and even recommend the professional.

Current evidence challenges the training of healthcare professionals, as the theoretical and technical training received seems insufficient. Moreover, ethical training does not necessarily guarantee good health practices, as the system also promotes actions that do not facilitate adequate care, which is difficult to overcome. One strategy is to focus on the interpersonal competencies of health professionals, for example, the incorporation of psychological concepts and elements of behavioral sciences [34], especially the development of cultural competencies [33] that allow professionals to provide care with cultural sensitivity and empathy in ways that favor positive care and improve health outcomes. Law No. 20.584 in Chile, the General Law of Health 2003 Ley General de Salud del año 2003, and the National Development Plan 2019–2024 in Mexico have proposed this; they encourage universal and effective access to health, prioritizing technical competence, medical quality, cultural relevance, and a non-discriminatory treatment [35]. Consequently, good health care should not be an exception but the rule in health systems.

One of the strengths of this study is that it provides a comprehensive approach to the study of culture and its relationship with health behaviors, comparing two Latino samples. Our statistical analyses allowed us to test structural invariance among Chileans and Mexicans and identified sources of cultural variability, in line with previous research [16,36]. However, the present study is also subject to some limitations, one of which is that we tested our hypothesis in a cross-sectional design; thus, we suggest caution in making causal inferences. Although mediation models can be tested in cross-sectional studies, a longitudinal model would allow for determination of causal relationships between variables [37]. In addition, we did not test the role of positive beliefs about physicians, which could coexist and moderate the proposed pattern of relationships. Finally, we could not measure actual behavior, although we used a valid self-reporting measure. In future studies, it will be interesting to examine actual behavior, for instance, manipulating interactions in an experimental study.

For future research, we propose inclusion of other types of variables, such as the health system or health insurance, to identify whether differences exist based on these factors, in addition to longitudinal studies that allow for follow-up and test changes in health behaviors.

## 5. Conclusions

Various models have been used to predict behaviors in health, some of which consider psychological variables and other that consider cultural factors. In this study, we used Betancourt’s model of culture, psychological processes, and behavior adapted for the study of health behavior in two samples, one Chilean and the other Mexican, to compare the models. The results show that the model is invariant in both samples, so negative beliefs about physicians have a direct effect on avoidance behaviors in health care. In addition, there is an effect mediated through psychological factors, such as perception of mistreatment and negative emotions associated with mistreatment.

These results are could account for the how culture influences health behaviors across countries, and they open the debate regarding how these variables can behave in other contexts.

## Figures and Tables

**Figure 1 ijerph-19-10067-f001:**
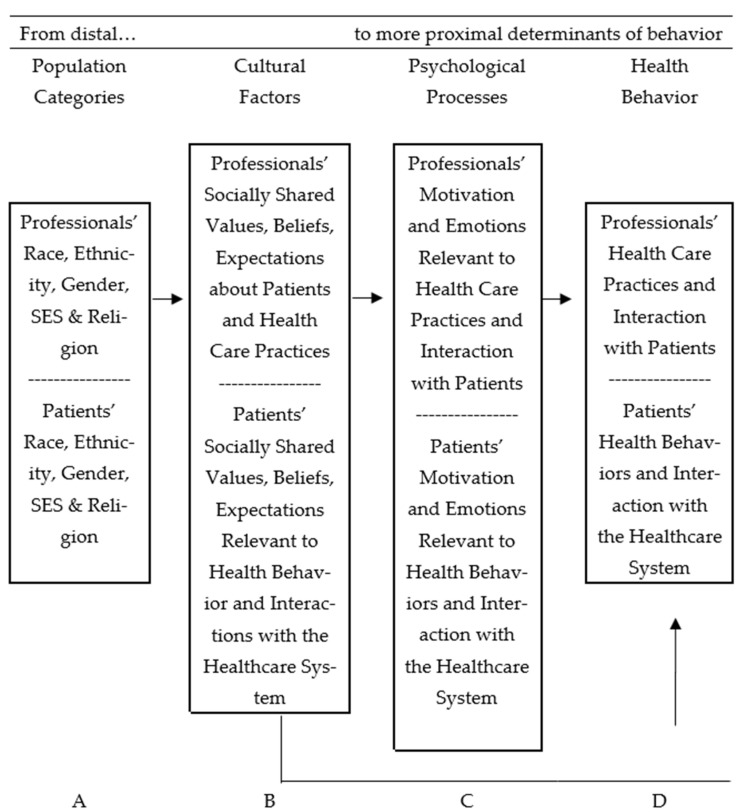
Betancourt’s integrative model of culture, psychology, and behavior was adapted to study health behavior [17,18,19].

**Figure 2 ijerph-19-10067-f002:**
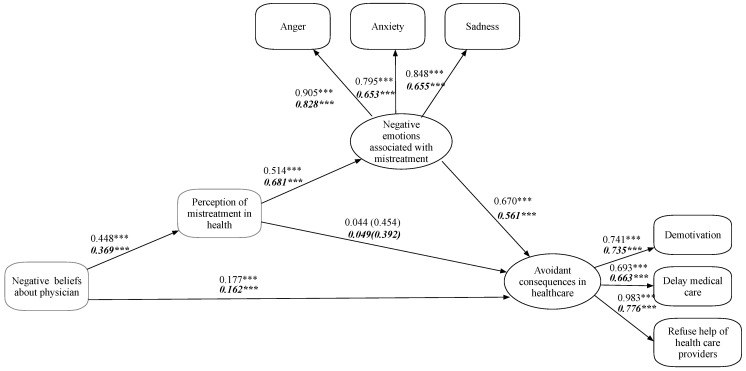
Own elaboration. Proposed model. Numbers in italics and bold correspond to the Mexican sample. *** *p* < 0.001.

**Table 1 ijerph-19-10067-t001:** Sociodemographic characteristics and comparison between groups.

Variable	Chile	Mexico	Test to Compare Groups	Effect Size
Sex			χ ^2^ = 37.346; *p* < 0.001	V = 0.238
Women	85.4%	64.9%		
Men	14.6%	35.1%		
Age	31.02 (7.46)	32.89 (14.63)	t = −2.044; *p* < 0.05	D = −0.12
Marital status			χ ^2^ = 50.427; *p* < 0.001	V = 0.27
Single	29.9%	50.5%		
Married	44.7%	33.5%		
Widow/widower	0.9%	2.8%		
Divorced	0.9%	3.4%		
Separated	1.8%	1.6%		
Living together	21.8%	8.2%		
Social class			χ ^2^ = 17.917; *p* < 0.01	V = 0.16
Low class	9.2%	3.2%		
Lower-middle class	19.7%	16.1%		
Middle class	51.7%	64.9%		
Upper-middle class	16.9%	14.9%		
Upper class	2.5%	0.9%		
Educational level			χ ^2^ = 21.238; *p* <0.001	V = 0.17
Less than high school (until 12 years old)	43%	56%		
University degree	47.8%	41.8%		
Postgraduate degree	9.2%	2.2%		

Source: Own elaboration.

**Table 2 ijerph-19-10067-t002:** Correlation matrix.

	Negative Beliefs about Physicians	Perception of Mistreatment in Health	Anger	Anxiety	Sadness	Demotivation	Delay Medical Care	Refuse Help of Healthcare Providers
Negative beliefs about physicians	-							
Perception of mistreatment in health	0.421 ***	-						
Anger	0.249 ***	0.494 ***	-					
Anxiety	0.187 ***	0.403 ***	0.615 ***	-				
Sadness	0.275 ***	0.473 ***	0.732 ***	0.636 ***	-			
Demotivation	0.307 ***	0.417 ***	0.425 ***	0.355 ***	0.405 ***	-		
Delay medical care	0.248 ***	0.402 ***	0.329 ***	0.286 ***	0.338 ***	0.676 ***	-	
Refuse help of healthcare providers	0.291 ***	0.403 ***	0.356 ***	0.299 ***	0.380 ***	0.577 ***	0.606 ***	-

Note: *** *p* < 0.001.

**Table 3 ijerph-19-10067-t003:** Summary of the models of configuration, measurement, and structural invariance analyzed according to country.

Step	Model	$\chi^{2}$	gl	*p*	RMSEA (IC)	CFI	TLI	SRMR	$\Delta \chi^{2}$ (SB)	∆ gl	∆CFI
1	Configural model	23.47	30	0.80	0.000 (0.00–0.04)	1.00	1.00	0.02	-	-	-
2	Measurement equivalence	32.39	34	0.55	0.000 (0.00–0.04)	1.00	1.00	0.03	8.92	4	0.00
3	Structural equivalence	51.63	35	<0.05	0.042 (0.01–0.07)	0.99	0.98	0.07	19.24	1	0.01

Source: Own elaboration.

## Data Availability

Not applicable.

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
