# Peer review of "Model to Predict Healthcare Behaviors: Comparison of a Chilean and Mexican Sample"

_ijerph, 2022, doi:10.3390/ijerph191610067_

Round 1

Reviewer 1 Report

The discussion and results should be more organized.  

Major corrections to the language are essential.

References should be matched with the instructions of this journal.

Author Response

Dear reviewer. We are thankful for your comments. We have taken them into consideration and improved our manuscript.

Comment #1: The discussion and results should be more organized.

Response: We have reorganized both sections. Changes were made directly into the manuscript.  

Comment #2:Major corrections to the language are essential

Response: We made a new revision. 

Comment #3:References should be matched with the instructions of this journal

Response: References are now matching the instructions.  

Reviewer 2 Report

The authors used multigroup structural equation modelling to test the structural relations among the cultural and psychological variables as determinant of healthcare avoidance behavior in Mexican and Chilean participants. The study fits the topic of IJERPH and the story is interesting. I recommend the acceptance after the following concerns are revolved.

1. There are some interesting finding in this study, i.e., the perception of mistreatment and negative emotions were less than intense for Chileans than Mexicans. The authors need to explain the findings, i.e., why negative emotions were less than intense for Chileans than Mexicans?

2. Line 121: ; changed to :

Author Response

Dear reviewer, we are thankful for your thoughtful review. We are positive that they guide us to submit a better and improved manuscript. Below we answered your observation one by one.

Comment #1:There are some interesting finding in this study, i.e., the perception of mistreatment and negative emotions were less than intense for Chileans than Mexicans. The authors need to explain the findings, i.e., why negative emotions were less than intense for Chileans than Mexicans?

Response: We added a response into the discussion section, under line 478-481.

Comment #2: Line 121: “,” changed to “:”

Response: It was modified in the document.

Reviewer 3 Report

Dear authors,

thank you for your interesting paper.

I have some questions:

1) Do you think that the answers are gender-specific?

2) Do you think you can transfer the model as is in other societies (democracy, dictatorship and so on)

3) People can change in stress situations like Covid. Can you use or adapt your model according to this?

Author Response

Dear reviewer. We highly appreciate your thoughtful comments. We are sure they helped us to improve our manuscript. 

Comment #1: Do you think that the answers are gender-specific?

Response: The pattern of relationships proposed can be gender-specific since gender can be considered a source of cultural variability. Nevertheless, we ran several t-test comparing men and females in all the variables of interest (e.g. negative belief about healthcare, perception of mistreatment; anxiety, demotivation, and so on); and we did not find any significant difference. Thus, in our study, we cannot conclude that the results are gender-specific. 

Comment #2: Do you think you can transfer the model as is in other societies (democracy, dictatorship, and so on).

Response: This comment is interesting, but it is out of the scope of our manuscript. We use Betancourt Integrative Model as theoretical background, a model that does not take into consideration contextual variables in a political system (democracy or dictatorship). Nevertheless, we do believe that living under a democracy or dictatorship can determine health policies, healthcare services, and health behaviors. 

Comment #3: People can change in stress situations like Covid. Can you use or adapt your model according to this?

Response: We agree that under stressful situations, like COVID-19 people may change their health behaviors. For instance, they might experience greater negative emotions, or face greater difficulties to adhere to treatments. The Betancourt model can be adapted considering these situations. 

Reviewer 4 Report

The paper explores whether the adherence to treatment and maintaining medical check-ups are important for health outcomes. However, these behaviors are not always carried out. Using Betancourt`s model for the study of health behavior; this study aims to examine the role of cultural beliefs about physicians, perceived mistreatment, and emotions associated with the experience of  mistreatment as antecedent of healthcare behavior among Chilean and Mexican primary care patients. This is a multivariate cross-sectional study with a non- probabilistic sample of 24 326 Mexican and 337 Chilean participants. Multigroup structural equation modelling was used to test the structural relations among the cultural and psychological variables as determinant of healthcare avoidance behavior. The results revealed the pattern of relationships which work in the same way for Chilean and Mexican samples. Negative cultural beliefs about physicians have a direct effect on avoidance behaviors for healthcare. In addition, there is an effect that is mediated through psychological factors such as perception of mistreatment and negative emotions associated to mistreatment. The structural invariance test shows that the perception of mistreatment and negative emotions were less than intense for Chileans than Mexicans. In contrast, the association between negative emotions and avoidance behaviors was less intense for Mexicans. The topic is interesting but there are some points to be addressed. The aim of the analysis should be evidenced in the abstract and introduction sections. The methodology should be further explained for replication. The conclusions should be improved with the weaknesses of the analysis and the in sights for future research. Finally, the manuscript should be English proofread because some sentences are not clear.

Author Response

Dear reviewer. We highly appreciate your observations. We went under a deep consideration of your comments and replied directly to the manuscript. 

Comment #1: The aim of the analysis should be evidenced in the abstract and introduction sections.

Response. The aim of the study is already declared in the abstract. For clarity purposes, we added it at the end of the introduction section, as well. 

Comment #2: The methodology should be further explained for replication.

Response: We agree with the comment. We added further details in the method section, procedure (Page 5, line 277).

Comment #3: The conclusions should be improved with the weaknesses of the analysis and the insights for future research.

Response: We agree. We made changes directly in the manuscript resubmitted

Round 2

Reviewer 1 Report

The discussion and results should be more organized.  

Major corrections for the language are essential.

References should be matched with the instructions of this journal.

Author Response

Dear Reviewer, 

We appreciate your comments so much, so we incorporated them into the new version of the manuscript.
Specifically, the results and discussion were more organized.
The manuscript was revised by proofreading.
And we correct the references.